# Conditional Loss of the Exocyst Component Exoc5 in Retinal Pigment Epithelium (RPE) Results in RPE Dysfunction, Photoreceptor Cell Degeneration, and Decreased Visual Function

**DOI:** 10.3390/ijms22105083

**Published:** 2021-05-11

**Authors:** Bärbel Rohrer, Manas R. Biswal, Elisabeth Obert, Yujing Dang, Yanhui Su, Xiaofeng Zuo, Ben Fogelgren, Altaf A. Kondkar, Glenn P. Lobo, Joshua H. Lipschutz

**Affiliations:** 1Department of Ophthalmology, Medical University of South Carolina, Charleston, SC 29425, USA; rohrer@musc.edu (B.R.); obert@musc.edu (E.O.); 2Ralph H. Johnson VA Medical Center, Division of Research, Charleston, SC 29401, USA; 3Department of Pharmaceutical Sciences, Taneja College of Pharmacy, University of South Florida, Tampa, FL 33612, USA; biswal@usf.edu; 4Department of Medicine, Medical University of South Carolina, Charleston, SC 29425, USA; dangy@musc.edu (Y.D.); su@musc.edu (Y.S.); zuo@musc.edu (X.Z.); lipschut@musc.edu (J.H.L.); 5Department of Anatomy, Biochemistry, and Physiology, University of Hawaii at Manoa, Honolulu, HI 96813, USA; fogelgre@hawaii.edu; 6Department of Ophthalmology, College of Medicine, King Saud University, Riyadh 11411, Saudi Arabia; akondkar@ksu.edu.sa; 7Department of Ophthalmology and Visual Neurosciences, Lions Research Building, 2001 6th Street SE., Room 225, University of Minnesota, Minneapolis, MN 55455, USA; 8Department of Medicine, Ralph H. Johnson Veterans Affairs Medical Center, Charleston, SC 29425, USA

**Keywords:** retinal pigmented epithelium, exocyst complex component 5, photoreceptor, visual function

## Abstract

To characterize the mechanisms by which the highly conserved exocyst trafficking complex regulates eye physiology in zebrafish and mice, we focused on Exoc5 (also known as *sec10*), a central exocyst component. We analyzed both *exoc5* zebrafish mutants and retinal pigmented epithelium (RPE)-specific *Exoc5* knockout mice. Exoc5 is present in both the non-pigmented epithelium of the ciliary body and in the RPE. In this study, we set out to establish an animal model to study the mechanisms underlying the ocular phenotype and to establish if loss of visual function is induced by postnatal RPE Exoc5-deficiency. *Exoc5*^−/−^ zebrafish had smaller eyes, with decreased number of melanocytes in the RPE and shorter photoreceptor outer segments. At 3.5 days post-fertilization, loss of rod and cone opsins were observed in zebrafish *exoc5* mutants. Mice with postnatal RPE-specific loss of Exoc5 showed retinal thinning associated with compromised visual function and loss of visual photoreceptor pigments. Abnormal levels of RPE65 together with a reduced c-wave amplitude indicate a dysfunctional RPE. The retinal phenotype in *Exoc5*^−/−^ mice was present at 20 weeks, but was more pronounced at 27 weeks, indicating progressive disease phenotype. We previously showed that the exocyst is necessary for photoreceptor ciliogenesis and retinal development. Here, we report that *exoc5* mutant zebrafish and mice with RPE-specific genetic ablation of Exoc5 develop abnormal RPE pigmentation, resulting in retinal cell dystrophy and loss of visual pigments associated with compromised vision. Together, these data suggest that exocyst-mediated signaling in the RPE is required for RPE structure and function, indirectly leading to photoreceptor degeneration.

## 1. Introduction

In developed countries such as the U.S., glaucoma and cataracts are treated relatively easily, so the most common cause of blindness is or soon will be retinal degenerative diseases [1]. Already, retinal diseases are the most common cause of childhood blindness worldwide [2]. In turn, almost one-quarter of known retinal degeneration genes in the Retinal Information Network (RetNet) are associated with ciliary function and intracellular trafficking [3]. Cilia, found on the surface of many eukaryotic cell types, are thin rod-like organelles extending outward from the basal body, a cellular organelle related to the centriole. Cilia are classified as primary (non-motile) or motile and contain a central axoneme composed of microtubules. In epithelia containing numerous motile cilia, cilia have a propulsive function, while primary cilia have a mechano- and chemosensory function [4]. It has been known for over 100 years that epithelial cells possess primary cilia; however, little had been ascribed regarding function, and primary cilia had even been considered to be vestigial [5]. Over the past decade, cilia have been shown to be central to the pathogenesis of “ciliopathies” that affect many organs, including the kidney and the eye. We discovered that the exocyst a highly conserved eight-protein trafficking complex, and its regulators are necessary for cilia formation in kidney and ocular cells [1,4,5,6,7,8,9]. In a series of studies, we elucidated the critical role that the highly conserved eight-protein exocyst trafficking complex plays in retinal and renal ciliogenesis [6,7,8,9,10].

Exocyst complex component 5 (Exoc5; also known as Sec10) is a central component of the exocyst complex [11]. The defect in eye development that we identified following antisense morpholino knockdown of *exoc5* in zebrafish, originally performed to study the role of the exocyst in renal ciliogenesis/development, was a serendipitous finding [12]. During the course of these studies, we noticed small eyes in morphant zebrafish and went on to demonstrate severe defects in cilia/eye development, following antisense morpholino knockdown of *exoc5* and the small GTPase *cdc42*, a regulator of the exocyst [12]. This led us to hypothesize that exocyst-mediated ciliogenic programs are conserved across organs and species and that the exocyst is centrally involved in mammalian eye development.

In the eye, photoreceptor and retinal pigmented epithelial (RPE) cells are ciliated. Mutations in exocyst proteins cause Joubert Syndrome [13], a ciliopathy associated with kidney and retinal phenotypes [14]. Our recently published data showed that two *exoc5* mutant zebrafish lines phenocopying *exoc5* morphants with grossly abnormal photoreceptor and RPE cells [15]. Using *Exoc5* floxed mice [16], together with *rhodopsin-Cre* mice, we generated *Exoc5* photoreceptor-specific knockout (KO) mice that, strikingly, showed progressive loss of photoreceptor cells and, by six weeks of age, were devoid of any visual response [15]. Overall, these data showed that exocyst-mediated ciliogenesis is conserved across species and organs. The progressive loss of the photoreceptor outer segments, which are modified primary cilia, in our zebrafish and mouse models was striking and led to blindness.

Given our previous results showing Exoc5 is necessary for ciliogenesis in photoreceptor cells in zebrafish and mice [15], and that cdc42 and exoc5 are necessary for retinal development in zebrafish and interact synergistically [12], here we investigated the role of Exoc5 in RPE cell structure and function and retinal development using RPE-specific *Bestrophin1-Cre* (*Best1-Cre^+^*) mice [17]. RPE-specific knockout of *Exoc5* led to RPE dysfunction retinal cell dystrophy as well as a loss of visual pigments in photoreceptors and was associated with compromised vision.

## 2. Results

### 2.1. Loss of *Exoc5* in Zebrafish Results in RPE and Photoreceptor Phenotypes

We obtained *exoc5*^+/−^ zebrafish lines (from the Zebrafish International Resource Consortium (ZIRC), *exoc5* #sa23168). At 3.5 days post-fertilization (dpf) we observed that all *exoc5*^−/−^ mutants had smaller eyes when compared to wild-type (WT) siblings [15]. The eye morphology phenotype was consistently observed in ~25% of the progeny from crosses of heterozygous parents, as would be expected for Mendelian inheritance of a recessive mutation. Detailed histological examination and H&E staining of 3.5 dpf *exoc5^−/−^* retinas showed thinning of the retinal pigmented epithelium (RPE) and photoreceptor layers (Figure 1A). By transmission electron microscopy (TEM), we confirmed shorter photoreceptor outer segments (OS), hypopigmentation, and fewer melanosomes in *exoc5^−/−^* retinas (Figure 1B,C). Such abnormalities were never observed in retinal sections of WT siblings. Taken together, these results indicate that eye development, proper retinal lamination, RPE health, and outer segment morphogenesis require Exoc5 function in zebrafish.

### 2.2. Loss of Exoc5 in Zebrafish Results in Early Retinal Phenotypes

We next performed a detailed immunohistochemical analysis in *exoc5^−/−^* zebrafish to determine if localization and trafficking of photoreceptor outer segment proteins occurred normally in *exoc5* mutants. At 3.5 dpf, ZPR3 localized normally to the rod outer segments of WT zebrafish (Figure 2A). In *exoc5* mutants, rudimentary outer segment localization of ZPR3 was observed (Figure 2A). To quantify the lengths of the outer segments, we used ZPR3 immunoreactivity as a surrogate, determining the extent of ZPR3 staining along the proximal-distal axis of the outer segments. In WT siblings, outer segments were 6.88 ± 0.28 μm in length (*n* = 25 embryos, 300 outer segments), while *exco5* mutant outer segments were 2.56 ± 0.12 μm in length (*p* < 0.001; *n* = 18 embryos, 200 outer segments) (Appendix A and Figure 2A’). Cone morphology, the predominant photoreceptor cell type in the zebrafish retina, was examined by immunolabeling with peanut agglutinin lectin (PNA-488), which labels the interphotoreceptor matrix surrounding cone outer segments and, to some extent, the retinal plexiform layers [15,18,19]. PNA-488 staining revealed that the *exoc5* mutant cone outer segments were significantly shorter (5.65 ± 0.30 μm in WT vs. 2.96 ± 0.25 μm in mutants; *p* < 0.001), disorganized and misshapen (Figure 2B,B’ and Appendix A). Additionally, immunohistological staining using R/G cone opsin antibody showed that the number of cone outer segments were significantly fewer in number and shorter in *exoc5* mutants compared to WT siblings (Figure 2C,C’), which suggested that systemic loss of *exoc5* results in defective cone outer segment morphogenesis. Heterozygous *exoc5**^+/−^* mutant carriers were then outcrossed for two generations with WT-Tg:XOPS-GFP [*Tg(XlRho:EGFP)^fl1^*], zebrafish line, which expresses soluble GFP under the control of rhodopsin in rod photoreceptor cells only [20]. The F3 generation heterozygous *exoc5* mutant carriers (*exoc5^+/−^*;Tg:XOPS-GFP) were crossed and retinas stained with R/G cone opsins. This analysis further confirmed loss of rhodopsin protein in rods and loss of cone opsins in retinas of exoc5^−/−^;Tg:XOPS-GFP mutants (Figure 3A–C). These data demonstrate that *exoc5* is indispensable for photoreceptor outer segment formation and maintenance in zebrafish. Because *exoc5^−/−^* zebrafish represented a global knockout of *exoc5*, it is unclear whether the loss of Exoc5 in photoreceptors and/or the RPE led to the phenotype. To answer this question, we generated a mouse model.

### 2.3. RPE-Specific Ablation of Exoc5 in Mice Cause Retinal Cell Degeneration

Both retinal pigmented epithelium (RPE) and photoreceptor cells are postmitotic in the healthy adult mammalian eye. Thus, interactions between specific photoreceptor outer segments (OS) and the adjacent RPE cells persist for life. Importantly, each RPE cell faces between 30 and 50 photoreceptor OS, and depending on its location in the retina, an individual RPE cell faces and functionally interacts with cones, rods, or (in most cases) a mixture of the two. To determine the role of Exoc5 in RPE for photoreceptor health in a mammalian model, Ai14 Cre reporter mice that harbor a *loxP*-flanked STOP cassette preventing transcription of a CAG promoter-driven red fluorescent protein variant (*td*Tomato), congenic on the C57BL/6J genetic background, were mated to *Exoc5fl/fl* mice, such that tdTomato expression in a cell indicates that *Exoc5* has been inactivated by Cre. To investigate the physiologic role of Exoc5 in the RPE, we produced mice with RPE-selective loss of EXOC5 in the early postnatal period by crossing mice bearing floxed *Exoc5* alleles (*tdTomatoExoc5**^flox/flox^*) with mice transgenic for *Best1*-*Cre^+^*. Deletion of *Exoc5* in RPE cells was confirmed by tdTomato expression (red color) in RPE cells of *tdTomatoExoc5fl/fl*;*Best1-Cre^+^*, compared to *tdTomatoExoc5fl/fl* mice (Appendix A).

We analyzed the retinas of mice at 20 and 27 weeks of age. Retinal histology of *Exoc5fl/fl*;*Best1-Cre^+^* mice at 20 weeks of age showed normal morphology when compared to controls (Figure 4A,A’). Quantitative analysis of ONL thickness and OS length revealed no difference at 20 weeks of age between the two genotypes (Figure 4C,C’). However, transmission electron microscopy (TEM) analysis suggested that OS in *Exoc5fl/fl*;*Best1-Cre^+^* mutants were ~10–15% shorter than those in age-matched littermate controls (*p* < 0.05) and also appeared thinner (Figure 4B,B’). At 27 weeks of age, *Exoc5fl/fl*;*Best1-Cre^+^* mice showed a significantly thinner outer nuclear layer (ONL) in histological sections (*p* < 0.05; Figure 4C, ONL layer thickness at 20 weeks versus Figure 5C, ONL layer thickness at 27 weeks), the photoreceptor OS appeared disorganized when compared to controls (*tdTomatoExoc5fl/fl*) mice (Figure 5A vs. Figure 5A’), and their length was significantly reduced (*p* < 0.001; OS lengths quantified in Figure 5C,C’). Disorganization and shorter photoreceptor outer segments (~73–80% shorter than controls; *p* < 0.005) were more obvious by transmission electron microscopy of conditional *Exoc5* knockout animals at 27 weeks (Figure 5B vs. Figure 5B’). While hypopigmentation and fewer melanosomes were identified in RPE from systemic *exoc5^−/−^* zebrafish, no apparent RPE phenotype could be detected in *Exoc5fl/fl*;*Best1-Cre^+^* mice at 27 weeks. Together, these data show the importance of EXOC5 expression in RPE cells for retinal cell maintenance.

### 2.4. Conditional Exoc5 Knockout Mice Show Defects in Photoreceptor Development

To determine if localization and trafficking of RPE and photoreceptor outer segment proteins occurred normally in *Exoc5fl/fl*;*Best1-Cre^+^* mice, we performed a detailed immunohistochemical analysis for RPE proteins and photoreceptor opsins. In both WT and *Exoc5fl/fl;Best1-Cre^+^* mice, rhodopsin was localized to the outer segments by 20 weeks of age; however, rhodopsin staining in *Exoc5 fl/fl;Best1-Cre^+^* mice was reduced to less than half (Figure 6A and quantified in Figure 6A’). At 27 weeks of age, rhodopsin levels in *Exoc5fl/fl;Best1-Cre^+^* mouse retinas was further decreased (Figure 7A and quantified in Figure 7A’). As we did for zebrafish retinas, rhodopsin immunoreactivity along the axis of the outer segment was used as a surrogate to assess outer segment length in mutant mice, revealing a shortening of rod OS (Appendix A). Cone morphology was examined by immunolabeling with an anti-Red/Green cone opsin antibody. Cone opsin staining revealed the presence of long and well-organized cone outer segments at 20 weeks for both WT and *Exoc5fl/fl;Best1-Cre^+^* mice, whereas *Exoc5fl/fl;Best1-Cre^+^* mouse cone outer segments at 27 weeks were significantly shorter and appeared disorganized and misshapen (Figure 6B and Figure 7B, Appendix A, cone opsins quantified in Figure 6B’ and Figure 7B’). Finally, the integrity of the retinal pigmented epithelium was accessed using RPE65 staining. This analysis showed a progressive decrease in RPE65 protein in retinas of *Exoc5fl/fl;Best1-Cre^+^* mice (Figure 6C and Figure 7C; quantified in Figure 6C’ and Figure 7C’). These results were supported by analysis of gene expression for RPE-specific proteins. mRNA levels for RPE65 and LRAT (Lecithin retinol acyltransferase), two enzymes crucial for the production of 11-*cis* retinal, were almost absent at 27 weeks in the *Exoc5**fl/fl;Best1-Cre^+^* mouse eyecups when compared to control. Conversely, levels for STRA6 (signaling receptor and transporter of all-*trans* retinol), a basally localized membrane receptor in the RPE critical for vitamin A uptake and homeostasis, was unaffected (Appendix A). These results demonstrate that loss of *Exoc5* in the RPE is crucial for RPE health and photoreceptor outer segment formation in mice.

### 2.5. Conditional Sec10 fl/fl, Best1-Cre^+^ KO Mice Show Significantly Reduced Rod Visual Function and RPE Integrity as Assessed by Electroretinography (ERG)

To correlate retinal structure with function, full-field ERG responses were analyzed, capturing ERG *a*- and *b*-waves under dark-adapted scotopic conditions to determine rod photoreceptor function, as well as *b*-waves under photopic conditions after light adaptation to determine cone function. In addition, *c*-waves were determined after a prolonged light flash to assess RPE integrity. Visual function was assessed in *Exoc5fl/fl* mice in the presence and absence of *Best1-Cre* at 6, 20, and 27 weeks of age. For ERGs with multiple light intensities, a repeated measure ANOVA followed by a post hoc ANOVA with Bonferroni correction was performed, individual amplitudes were compared by *t*-test.

Within the wild-type/control group, an amplitude by genotype interaction was identified for *b*- but not *a*-wave amplitudes (*a*-waves, *p* = 0.15; *b*-waves, *p* < 0.02), which was driven by larger *b*-wave amplitudes at 20 weeks of age. In the *Exoc5fl/fl;Best1-Cre+* mice an amplitude by genotype interaction was also identified for *b*- but not *a*-wave amplitudes (*a*-waves, *p* = 0.33; *b*-waves, *p* < 0.03), which was driven by smaller *b*-wave amplitudes at 27 weeks of age (*p* = 0.003). Cone amplitudes and *c*-wave amplitudes were unaffected by age in control mice, whereas in *Exoc5fl/fl;Best1-Cre^+^* mice, cone amplitudes remained stable, but *c*-wave amplitudes dropped with age (*p* = 0.025; 6 versus 27-weeks: *p* = 0.002) (Figure 8 and Appendix A).

When the amplitudes were analyzed within the age groups, but differentiated by genotype, the following conclusions could be drawn. At 6 weeks of age, there was no difference in visual function and RPE integrity between the control and *Exoc5fl/fl;Best1-Cre^+^* mice (*a*-waves: *p* = 0.85, *b*-waves: *p* = 0.86, cones: *p* = 0.71, *c*-waves: *p* = 0.90). At 20 weeks of age, a difference in rod visual function and RPE integrity between the control and *Exoc5fl/fl;Best1-Cre^+^* mice was identified (*a*-waves: *p* = 0.003, *b*-waves: *p* = 0.0001, cones: *p* = 0.45, *c*-waves: *p* = 0.024). Likewise, at 27 weeks of age, a difference in rod visual function and RPE integrity between the control and *Exoc5fl/fl;Best1-Cre^+^* mice was identified (*a*-waves: *p* = 0.02, *b*-waves: *p* = 0.0006, cones: *p* = 0.41, *c*-waves: *p* = 0.020) (Figure 8 and Appendix A).

In summary, eliminating Exoc5 from RPE impaired RPE *c*-wave amplitudes with age concomitant with a reduction in signals in the rod pathway, whereas cone function was unaffected.

## 3. Discussion

We report two principal findings here, both of which are of great interest. First, we show that the exocyst, which we previously showed is necessary for normal photoreceptor development and function [15], is also required for normal function and maintenance of the RPE. A strength of this paper is that we have dissected out RPE maintenance as the variable that we are studying. The RPE and the eye were able to develop normally as Cre expression in the *Best1-Cre* mouse first occurs at ~P15 and is maximal by P28/4 weeks [17]. By examining 20 and 27-weeks eyes from *Exoc5* RPE-specific KO mice, we show that the degeneration of the RPE is progressive and occurs after the conditional loss of Exoc5. In the eye, photoreceptors and RPE cells are ciliated. We have previously shown that the exocyst is necessary for ciliogenesis in the kidney [10], eye [15], ear [21], and heart [22]. Retinal degeneration is the most common phenotype among ciliopathy patients. Most research on the retinal phenotype has focused on retinal photoreceptors that contain a highly specialized primary cilium that is found to degenerate in ciliopathy animal models and patients [23,24,25,26]. The contribution of other ciliated retinal cell types to retinal degeneration has not been investigated so far.

The RPE is a ciliated monolayer epithelium that lies at the back of the eye and is essential for photoreceptor development and function. We hypothesize that loss of cilia function in the RPE cells is what is leading to loss of function of the RPE, which leads to progressive photoreceptor cell degeneration in our mouse model. Second, normal photoreceptor function depends on normal RPE function. RPE-photoreceptor interactions and interdependence have been known for many years, and the roles of the RPE in visual function are summarized [27]. First, the RPE is required for photoreceptor disc shedding and lack of photoreceptor outer segment phagocytosis by the RPE, whether due to genetic defects in RPE (Mertk, αvβ5 Integrin Receptor) or in photoreceptors (MFG-E8), results in photoreceptor degeneration [28]. The presence of rod OS packages in the subretinal space of *Exoc5**fl/fl;Best1-Cre^+^* mouse retina suggests that OS phagocytosis might be impaired, although phagocytosis assays have not yet been performed. Second, the RPE is required for retinoid synthesis, and the lack of 11-*cis* retinal in RPE65^−/−^ mice results in slow rod [29] and rapid cone degeneration [30] associated with cone opsin but not rhodopsin mislocalization, and both rod and cone ERGs are affected in these mice. This retinoid-dependent phenotype is replicated in LRAT^−/−^ mice [30]. The reduced RPE65 protein staining, together with the significant reduction in mRNA for RPE65 and LRAT, suggest that the rod and cone phenotype identified in *Exoc5**fl/fl;Best1-Cre^+^* mice is associated with impaired levels of the chromophore 11-*cis* retinal (Appendix A). Third, and a mechanism not yet explored, is the interdependence of the energy metabolism between RPE and photoreceptors. RPE cells deliver glucose to photoreceptors for aerobic glycolysis and, in return, obtain lacate from photoreceptors to generate energy through oxidative phosphorylation [31]. An imbalance in this metabolic relationship leads to degeneration of either of the two cell types. Damaged RPE cells might increase glycose use, leading to metabolic dysfunction in photoreceptors followed by degeneration [32].

We propose another speculative hypothesis. Two groups had shown that the primary cilia have been shown to secrete small extracellular vesicles (termed ectosomes) in Chlamydomonas [23] and Caenorhabditis elegans [24] and are responsible for up to 60% of the small (50–150 nm) extracellular vesicles (EVs) that are produced and released from cells [25]. One group showed in mammalian cells that, when activated, G protein-coupled receptors fail to undergo retrieval from cilia back into the cell. These G protein-coupled receptors concentrate into membranous buds at the tips of cilia before being released via ectosomes, and hedgehog-dependent ectocytosis regulates ciliary signaling. We recently showed in mammalian cells that not only do primary cilia secrete ectosomes but that primary cilia are responsible for up to 60% of the small (50–150 nm) extracellular vesicles (EVs) that are secreted [25]. In RPE monolayers, we have shown that RPE cells release EVs toward both the apical and basal side and that these vesicles are taken up by RPE target cells, resulting in changes in cell behavior [33,34,35]. Photoreceptor cilia have the ability to produce ectosomes, which under normal conditions contribute to outer segment disk formation, but in the absence of rds/peripherin are released as ectosomes [36]. Interestingly, small vesicles reminiscent of ecotosomes observed in the rds^−/−^ mouse rods [36] were visible at the base of the poorly formed fish photoreceptor outer segments (Figure 1C). We hypothesize it is plausible that there is crosstalk between the RPE and photoreceptors that is mediated by EVs potentially going in both directions, and this hypothesis warrants further investigation. However, as the exocyst localizes in other structures besides the primary cilium and contributes to the modulation of synthesis and delivery of secretory and basolateral plasma membrane proteins, it is impossible to attribute the retinal and RPE phenotype solely to a ciliary defect caused by the loss of Exoc5. We have generated intraflagellar transport protein 88 (IFT88) fl/fl mice, which will be crossed with *Best1-Cre^+^* mice, to support the idea that primary cilia defects in RPE lead to the RPE and photoreceptor phenotype observed here, as Ift88 is uniquely localized to the cilia basal body and the cilia axoneme and is required for the assembly and function of the primary cilia [37].

Taken together, our results not only provide insights into Exoc5 function and dysfunction but also suggest the existence of a molecular network that includes Exoc5 function and the RPE/Photoreceptor crosstalk, opening a new perspective in using this knowledge toward the development of novel therapeutic strategies for the treatment of inherited retinal dystrophies.

## 4. Materials and Methods

### 4.1. Materials

All chemicals, unless stated otherwise, were purchased from Sigma-Aldrich (St. Louis, MO, USA).

### 4.2. Animal Use Approval

All experiments on zebrafish and mice were approved by the Institutional Animal Care and Use Committee (IACUC Protocol#0078) of the Medical University of South Carolina and the Ralph H. Johnson VAMC and were performed in compliance with the ARVO Statement for the Use of Animals in Ophthalmic and Vision Research.

### 4.3. Zebrafish Husbandry

Adult zebrafish were maintained and raised in an Aquatic Habitats recirculating water system (Tecniplast, West Chester, PA) in a 14:10-h light-dark cycle and maintained under standard conditions at 28.5 °C. The *exoc5* mutant zebrafish line was purchased from Zebrafish International Resource Center (ZIRC, Oregon, OR: *exoc5*-sa23168). This exoc5 zebrafish mutant contains a C>T point mutation at amino acid 377, resulting in a premature stop codon. The point mutation was verified by PCR and direct sequencing of both strands in both heterozygote adults and mutant larvae progeny. The transgenic *Tg(XlRho:EGFP)^fl1^*, WT (strain AB/TU) zebrafish lines have been described previously and were crossed with *exoc5*^+/−^ animals to finally generate *exoc5^+/−^*;Tg-XOPS-GFP breeding pairs. The Tg:XOPS-GFP zebrafish line expresses soluble GFP driven by rhodopsin in rod photoreceptor cells only (Fadool, 2003; Perkins and Fadool, 2010). Collected embryos were maintained in embryo medium (15 mM NaCl, 0.5 mM KCl, 1 mM CaCl_2_, 1 mM MgSO_4_, 0.15 mM KH_2_PO_4_, 0.05 mM NH_2_PO_4_, 0.7 mM NaHCO_3_) at 28.5 °C. Morphological features were used to determine the stage of the embryos in hours (hpf) or days (dpf) post-fertilization. Genomic DNA from clipped fins or whole 3.5 days post fertilized (dpf) zebrafish after phenotypes were observed were extracted in 50 μL 1× lysis buffer (10 mM Tris-HCl pH 8.0, 50 mM KCl, 0.3% Tween 20, 0.3% NP40), denatured at 98 °C for 10 min, digested at 55 °C for 6 h after 10 μg/mL proteinase K was added, and the reaction was stopped at 98 °C for 10 min. The PCR primers used for genotyping were forward primer, 5′-CTATATAGACATGGAGCGGCAAT-3′; reverse primer: 5′-CCAACAATTCCTCACCTTCC-3′. Sequencing was performed by Genewiz with the forward primer used for PCR (Genewiz, South Plainfield, NJ, USA).

### 4.4. Mouse Husbandry

Animals were kept in a 12-h light-dark cycle with food and water *ad libitum*. The generation and genotyping of our *Exoc5*(Sec10)fl/fl mice have been described previously [16]. The RPE-specific conditional *Exoc5* knockout mice were generated by crossing *Exoc5* fl/fl with *Best1-Cre^+^* mice (Jackson Laboratories, Bar Harbor, ME, USA) and are designated as *Exoc5fl/fl;Best1-Cre^+^* in the manuscript (Appendix A). *Exoc5* conditional mice were genotyped using the following set of PCR primers: forward primer, Fl-Exoc5 5′loxP#2F 5′-GCCTGTAACTCACAGAGATC-3′ with reverse primer, Fl-Exoc5 5′loxP#2R 5′-GCTGGCATTCTAAGTCATGG-3′, tDTomato: forward primer, 5’-CTCTGCTGCCTCCTGGCTTCTR-3′ with reverse primer, 5’-TCAATGGGCGGGGGTCGTT-3′, *Best1-Cre^+^* mice were identified using the following set of PCR primers: forward primer, 5′ Cre-F:5’-TTGCCTGCATTACCGGTCGATGCAACGAGT-3′; reverse primer: Cre-R 5′-CCTGGTCGAAATCAGTGCGTTCGAACGCTA-3′.

### 4.5. Mouse Retina Dissection, Immunohistochemistry, and Fluorescence Imaging

Whole zebrafish or mouse eyes were enucleated and fixed by immersion in 4% paraformaldehyde in 1× phosphate buffer (PBS) for 2 h at room temperature (RT). Eyes were incubated in a sucrose gradient of 5% sucrose in phosphate buffer (SPB) for 15 min at RT, 15% SPB for 15 min at RT, 30% SPB for 2 hours at RT, overnight in a 70:30 *v/v* ratio of OCT:SPB solution (Tissue-Tek, Sakura Finetech, Torrance, CA, USA) at 4 °C. Eyes were then mounted in cyro-molds containing 70:30 *v*/*v* ratio of OCT:SPB solution and frozen on a dry-ice bath containing 100% ethanol. A total of 12 μm thick sections were cut using a cryostat (Leica, Welzlar, Germany). Sections were air-dried for 24 h at RT and then subjected to immunohistochemistry. Blocking solution (1% BSA, 5% normal goat serum, 0.2% Triton-X-100, 0.1% Tween-20 in PBS) was applied for 2 h in a humidified chamber. Primary antibodies were diluted in blocking solution as follows: ZPR3 (zebrafish eyes only; 1:100 dilution; ZIRC), 1D4-rhodopsin (for mouse eyes only; Sigma, St. Louis, MO), RPE65 (1:200 dilution; Sigma), and acetylated-α-tubulin (1:1000 dilution; Sigma, St. Louis, MO). Medium-wavelength cones were stained with R/G cone opsins (1:250 dilution, Molecular Probes, Eugene, OR). Conjugated PNA-488 antibody was purchased from Invitrogen Life Technologies (Carlsbad, CA, USA) and used at 1:500 dilution, and 4′,6-diamidino-2-phenylendole (DAPI; 1:10,000) was used to label nuclei. Sections were mounted in Vectashield (Vector Laboratories, Burlingame, CA, USA). Z-stack images were collected using a Leica SP8 confocal microscope (Leica, Welzlar, Germany) and processed with the Leica Viewer software.

### 4.6. Transmission Electron Microscopy (TEM)

Control and *exoc5* mutant zebrafish larvae or WT and *Exoc5fl/fl;Best1-Cre^+^* mice eyes (or eyecups) were fixed in a solution containing 2.5% glutaraldehyde, 2% paraformaldehyde, and postfixed with 2% osmium tetroxide. The fixed tissue was sectioned to obtain radial sections at 1 μm and rinsed with cacodylate buffer (0.1 M), dehydrated through a graded ethanol series, and infiltrated with Epon resin. Samples were processed by the Electron Microscopy Resource Laboratory at the Medical University of South Carolina (MUSC) or the University of South Florida (USF) using a Joel Transmission Electron Microscope (JEM-1400Plus, Peabody, MA).

### 4.7. Electroretinography (ERG)

Electroretinography (ERGs) recordings on mice were performed according to published procedures [18]. Mice were dark-adapted overnight, anesthetized using xylazine (20 mg/kg) and ketamine (100 mg/kg), and pupils were dilated with 1 drop each of phenylephrine HCl (2.5%) and tropicamide (1%). Body temperature was stabilized via a DC-powered heating pad held at 37 °C. ERG recordings and data analyses were performed using the EPIC-4000 system (LKC Technologies, Inc., Gaithersburg, MD), using light stimuli with varying light intensities and wavelengths. Under scotopic conditions, responses to 10 μs single-flashes of white light (maximum intensity of 2.48 photopic cd–s/m^2^) between 40 and 0 decibels (dB) of attenuation were measured. After light-adapting animals for 8 min with rod-saturating light [35], UV-cone responses were tested using LED flashes centered at 360 nm at a single light intensity. Peak *a*-wave amplitude was measured from baseline to the initial negative-going voltage, whereas peak *b*-wave amplitude was measured from the trough of the *a*-wave to the peak of the positive *b*-wave.

### 4.8. Statistics

Results are presented as mean ± standard deviation for image analysis, mean ± standard error of the mean for electroretinography. For pairwise comparisons, statistical significance was assessed using the two-tailed Student’s *t*-test. For ERG experiments, testing from the same animals over multiple light intensities, repeated measure ANOVA followed by Fisher’s PLSD was used (StatView). For Western blot analysis, relative intensities of each band were quantified (densitometry) using ImageJ Software version 1.49 and normalized to the loading control.

## 5. Conclusions

Our results show the pathological consequences of condition Exoc5 protein loss in RPE on retinal cell function. Loss of Exoc5 in RPE resulted in progressive retinal cell degeneration and loss of visual function. Further studies using IFT88 mice are needed to confirm if loss of cilia results in these retinal phenotypes.

## Figures and Tables

**Figure 1 ijms-22-05083-f001:**
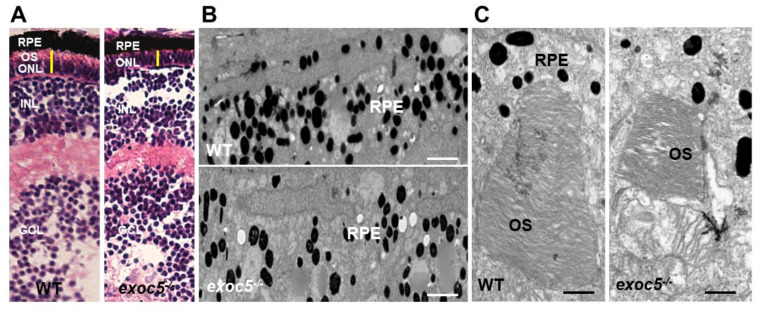
Histological and TEM analysis of RPE and retinas of wild-type and *exoc5* mutant zebrafish larvae. (**A**) In systemic *exoc5* homozygous mutants (*exoc5**^−/−^*), the photoreceptor outer segments (OS) were shorter compared to wild-type (WT) siblings. (**B**) Ultrastructural analysis of WT and *exoc5**^−/−^* mutant RPE using transmission electron microscopy indicate reduced levels of melanosomes in the mutant fish RPE. (**C**) WT photoreceptors showed tightly stacked disk membranes, while in *exoc5**^−/−^* mutants, only remnants of outer segments (OS) could be observed. Please note the apparent formation of ectosomes at the base of the mutant photoreceptor OS. Scale bars = 2 μm (**B**), 800 nm (**C**). OS, outer segments; ONL, outer nuclear layer; RPE, retinal pigmented epithelium; INL, inner nuclear layer; GCL, ganglion cell layer.

**Figure 2 ijms-22-05083-f002:**
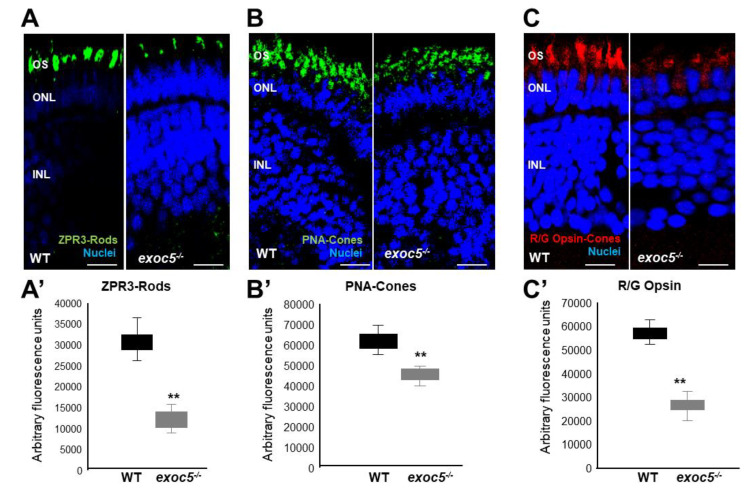
Immunohistochemical analysis of rod and cone photoreceptors in wild-type and *exoc5^−/−^* mutant zebrafish. Rod photoreceptor outer segments were identified with ZPR3 antibody (green, ZPR3, (**A**)), cone photoreceptors with PNA-lectin-488 (green, PNA, (**B**)), and medium-wavelength R/G cone opsins (red, R/G Opsin, (**C**)), all at 3.5 dpf. Loss of rod and cone OS immunofluorescence was noted in *exoc5^−/−^* mutant zebrafish. Scale bars = 50 μm (**B**) and 25 μm (**A**,**C**). OS, outer segments; ONL, outer nuclear layer; INL, inner nuclear layer. (**A’**–**C’**) Image *J* was used to quantify immunofluorescence for the 3 OS markers. ** *p* < 0.005 (WT vs. *exoc5^−/−^* mutants).

**Figure 3 ijms-22-05083-f003:**
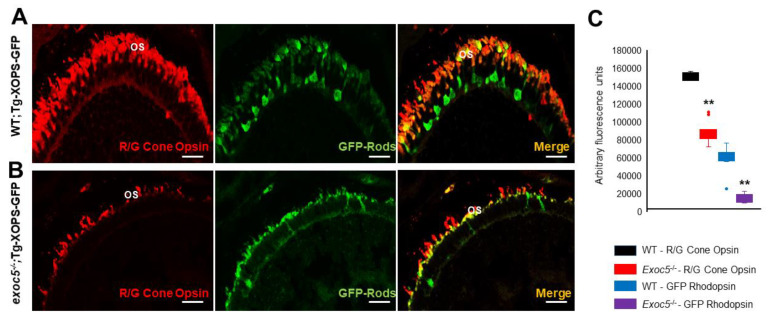
Immunohistochemical analysis of rod and cone photoreceptors in transgenic (Tg:XOPS) wild-type and *exoc5* mutant zebrafish. Top panel: wild-type:Transgenic-XOPS (WT;Tg-XOPS-GFP) zebrafish retinas immunostained with R/G cone opsin (red). Bottom panel: *exoc5^−/−^* mutant:Transgenic-XOPS (*exoc5^−/−^*;Tg-XOPS-GFP) retinas immunostained with R/G cone opsin (red). The transgenic line expresses soluble endogenous GFP in rods (green). Significantly shorter rod and cone OS were observed in *exoc5^−/−^* mutant zebrafish, compared to WT zebrafish, at similar ages. (**C**) Image *J* was used to quantify fluorescence for R/G cone opsin and GFP+ rods. ** *p* < 0.005 (WT vs. *exoc5^−/−^* mutants).

**Figure 4 ijms-22-05083-f004:**
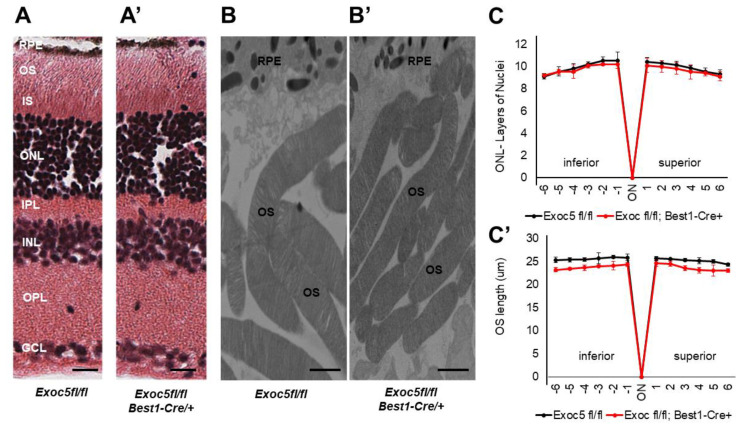
Histological and TEM analysis of retinas of *Exoc5^−/−^* mutants and *WT* mice at 20 weeks of age. (**A**,**A’**) H&E staining of WT and *Exoc5^−/−^* retinas at 20 weeks did not reveal histological differences. (**B**,**B’**) Transmission electron microscopy of wild-type (WT) mice photoreceptor cells show that rod OSs are shorter and thinner in *Exoc5^−/−^* retinas. The thickness of the ONL (**C**) and OS lengths (**C’**) from H&E sections through the optic nerve (ON; 0 μm distance from Optic Nerve and the starting point) was measured at 12 locations around the retina, six each in the superior and inferior hemispheres, each equally at 150 μm distances. (**B,B’**) Scale bars = 800 nm. OS, outer segments; RPE, retinal pigmented epithelium; IS, inner segments; ONL, outer nuclear layer; INL, inner nuclear layer; OPL, outer plexiform layer.

**Figure 5 ijms-22-05083-f005:**
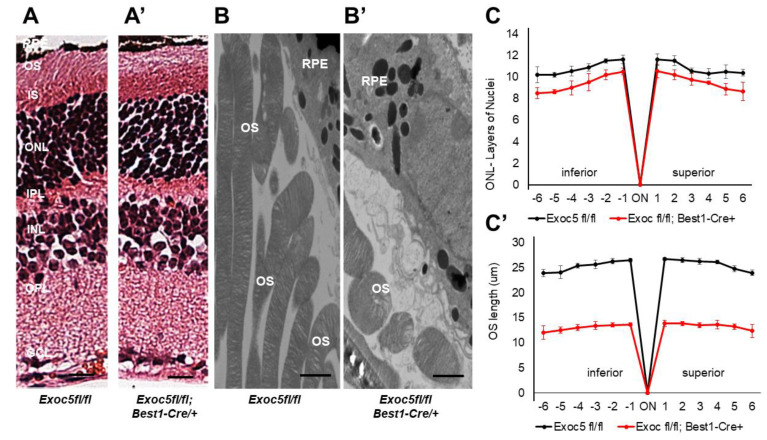
Histological and TEM analysis of retinas of *Exoc5^−/−^* mutants and *WT* mice at 27 weeks of age. (**A,A’**) H&E staining of WT and *Exoc5^−/−^* retinas at 27 weeks suggests the presence of disorganized rod outer segments (OS). (**B,B’**) Transmission electron microscopy of WT and *Exoc5^−/−^* photoreceptor cells reveal packages of rod OS in the subretinal space. (**C**,**C’**) Thickness of the ONL (**C**) and OS lengths (**C’**) from H&E sections through the optic nerve (ON; 0 μm distance from Optic Nerve and starting point) was measured at 12 locations around the retina, six each in the superior and inferior hemispheres, each equally at 150 μm distances. Scale bars = 800 nm (**B,B’**). OS, outer segments; RPE, retinal pigmented epithelium; IS, inner segments; ONL, outer nuclear layer; INL, inner nuclear layer; OPL, outer plexiform layer.

**Figure 6 ijms-22-05083-f006:**
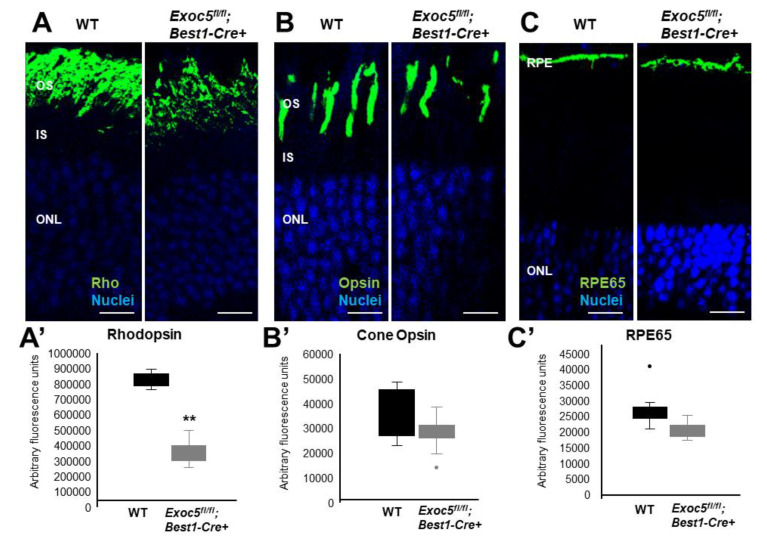
Immunohistochemical analysis of rod and cone photoreceptors in wild-type and conditional *Exoc5* knockout mice at 20 weeks of age. Levels and localization of rhodopsin (green, Rho, (**A**)), red/green cone opsins (green, R/G Opsin, (**B**)), and RPE65 (green, RPE65, (**C**)) were assessed using immunohistochemistry in retinal sections of 20-weeks-old mice to identify alterations in rod and cone outer segments, as well as retinal pigment epithelium (RPE). Only moderate differences in staining were observed. Scale bars = 50 μm (**A**–**C**). Image *J* was used to quantify rhodopsin (**A’**) and cone opsin (**B’**) fluorescence in photoreceptors, and RPE65 (**C’**) in RPE of WT and *Exoc5fl/fl;Best1-Cre^+^* mice. OS, outer segments; IS, inner segments; ONL, outer nuclear layer, INL, inner nuclear layer; RPE, retinal pigmented epithelium. ** *p* < 0.05 (WT vs. *Exoc5fl/fl;Best1-Cre^+^* mutants).

**Figure 7 ijms-22-05083-f007:**
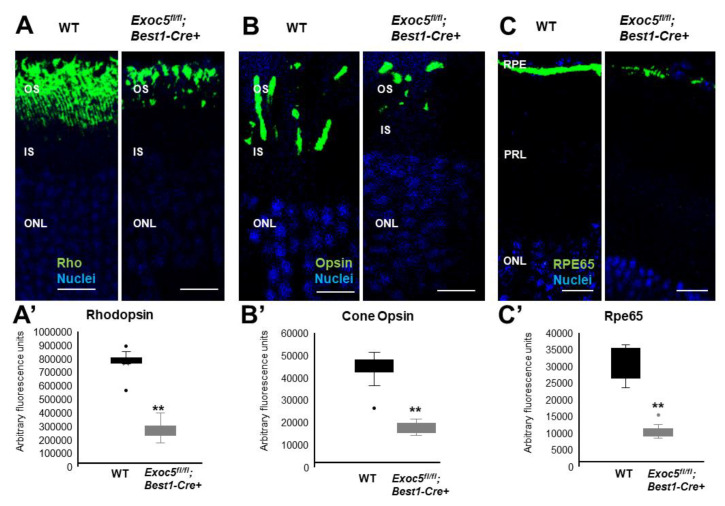
Immunohistochemical analysis of rod and cone photoreceptors in wild-type and conditional *Exoc5* knockout mice at 27 weeks of age. Levels and localization of rhodopsin (green, Rho, (**A**)), Red/Green cone opsins (green, R/G Opsin, (**B**)), and RPE65 (green, RPE65, (**C**)), were assessed using immunohistochemistry in retinal sections from mice collected at 27 weeks of age to identify alterations in rod and cone outer segments and RPE. Rod and cone outer segments are dysmorphic, and RPE staining for RPE65 was severely disrupted. Image *J* was used to quantify rhodopsin (**A’**) and cone opsin (**B’**) fluorescence in photoreceptors, and RPE65 (**C’**) in RPE of WT and *Exoc5fl/fl;Best1-Cre^+^* mice. Scale bars = 50 μm (**A**–**C**). OS, outer segments; IS, inner segments; ONL, outer nuclear layer, INL, inner nuclear layer; RPE, retinal pigmented epithelium. ** *p* < 0.05 (WT vs. *Exoc5fl/fl;Best1-Cre^+^* mutants).

**Figure 8 ijms-22-05083-f008:**
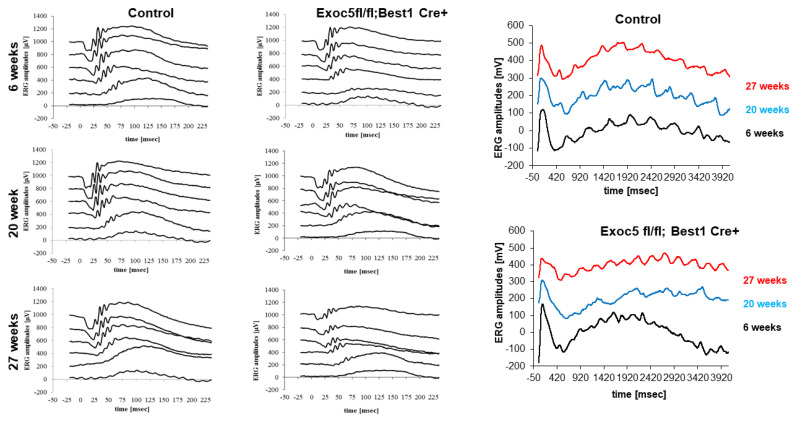
Measurement of visual function in *Exoc5fl/fl;Best1-Cre^+^* mice by full-field electroretinography (ERG). Dark-adapted, scotopic ERGs were recorded in response to increasing light intensities in cohorts of 6, 20, and 27-week old control (*Exoc5fl/fl*) and *Exoc5fl/fl;Best1-Cre^+^* mice. RPE-specific Exoc5 knockout mice in which both copies of Exoc5 were eliminated showed progressively significantly lower dark-adapted *a*- and *b*-wave amplitudes compared to controls (posthoc ANOVA: *a*-waves *p* < 0.03; *b*-waves *p* < 0.01), in particular at higher light intensities (20, 10, 6, 0 dB). *C*-waves showed an age-dependent decline in amplitude in the *Exoc5 fl/fl;Best1-Cre^+^* mice (*p* = 0.002). Data are expressed as mean ± SEM (*Exoc5 fl/fl;Best1-Cre^-^*: *n* = 10; and *Exoc5 fl/fl;Best1-Cre^+^* mice: *n* = 10).

## Data Availability

Not applicable.

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
