# Peer review of "Conditional Loss of the Exocyst Component Exoc5 in Retinal Pigment Epithelium (RPE) Results in RPE Dysfunction, Photoreceptor Cell Degeneration, and Decreased Visual Function"

_ijms, 2021, doi:10.3390/ijms22105083_

Round 1

Reviewer 1 Report

In prevoius published experiments the authors had investigated exoc5 ko zebrafish lines and found that PRE and photoreceptor cells were both affected. In a knock-out mouse model exoc5 was deleted in photorector cells resulted in blindness of mice. In the current experiments celltype specific knockout in PRE cells also affected photoreceptor cells indirectly. The authors demonstrate these findings by histological, EM examinations and physiological experiments with the mice.

Comment:

1) The findings are mainly descriptive. A biological explanation, e.g. reduction of extravesicular vesicles secreted by ko PRE cells was not investigated as well as the effect of substitution of vesicals from normal eyes.

2) The hedgehog pathway dependent secretion of vesicles from cilia in exoc5 ko and wt PRE cells would have been interesting.

The authors should at least comment on these issues.

Author Response

Reviewer 1

In prevoius published experiments the authors had investigated exoc5 ko zebrafish lines and found that PRE and photoreceptor cells were both affected. In a knock-out mouse model exoc5 was deleted in photorector cells resulted in blindness of mice. In the current experiments celltype specific knockout in PRE cells also affected photoreceptor cells indirectly. The authors demonstrate these findings by histological, EM examinations and physiological experiments with the mice.

Comment:

1) The findings are mainly descriptive. A biological explanation, e.g. reduction of extravesicular vesicles secreted by ko PRE cells was not investigated as well as the effect of substitution of vesicals from normal eyes.

2) The hedgehog pathway dependent secretion of vesicles from cilia in exoc5 ko and wt PRE cells would have been interesting.

The authors should at least comment on these issues.

Author response: We thank the reviewer for his/her comments. We have disused these in detail in our revised manuscript (lines 330-377). We are currently investigating extracellular vesicle (EV) release among genotypes. As suggested by the reviewer in follow-up studies we will also investigate the hedgehog pathway in EV secretion among the WT and KO lines.

Reviewer 2 Report

Experiments are very well done, data are convincing and conclusions are sound. Although additional experiments are always possible, data provided convey a strong message. 

Describe the long-term consequences of RPE-specific ablation of Exoc5 in mice. Is neovascularization ever observed in the mutants? 

Are any changes observed in RPE/choriocapillaris?

Provide more details regarding the Best1-Cre line, especially regarding % recombination. 

In Figure 2, 3, 6, 7, and 8 (and RPE/choriocapillaris), the authors should quantify and analyse statistically the changes in the retina.

In Figure 4 and 5, the authors should measure the thickness of retinal layers. Are there any significant differences in thickness of ONL and OS ?

Author Response

Reviewer 2

Comments and Suggestions for Authors

Experiments are very well done, data are convincing and conclusions are sound. Although additional experiments are always possible, data provided convey a strong message. 

Author Response: We thank the reviewer for his/her comments and for the support of our study.

 Describe the long-term consequences of RPE-specific ablation of Exoc5 in mice. Is neovascularization ever observed in the mutants? 

Author Response: We have addressed this in our discussion (lines 330-377). We have not tested for neovascularization, but we have plans do so in our follow up study.

Are any changes observed in RPE/choriocapillaris?

Author Response: We have not tested the mice for changes in choriocapillaris, but we will do so in our follow up study.

 Provide more details regarding the Best1-Cre line, especially regarding % recombination.

Author Response: The Best1 cre+ line was originally generated by our collaborator, Dr. Josh Dunaief at Penn, who deposited these mice at the JAX labs. We obtained these mice from the JAX labs. These mice express Cre recombinase under the control of the human bestrophin 1 (BEST) promoter. This strain represents an effective tool for generating retinal pigment epithelium (RPE) specific-targeted mutants that would be useful in studies of human retinopathies. The Best-1Cre+ mice were crossed with the tDTomato-Exoc5fl/fl line to generate (tDTomato-Exoc5fl/fl; Best1 Cre+) mice. Loss of Exoc5 in the RPE was more than 98% (% recombination) based on tDTomato positive expression in the RPE cells, such that tdTomato expression in a cell indicates that Exoc5 has been knockout by Cre (lines 164-173 and Suppl. Fig. S1).

In Figure 2, 3, 6, 7, and 8 (and RPE/choriocapillaris), the authors should quantify and analyse statistically the changes in the retina.

Author Response: We have quantified retinal changes in figures 2, 3, 6, 7 and 8. We show these in the revised manuscript, using fluorescence quantification (Pages 4, 5, 8 and 9).

In Figure 4 and 5, the authors should measure the thickness of retinal layers. Are there any significant differences in thickness of ONL and OS ?

Author Response: we have performed measurement of ONL and OS lengths using H&E sections and these are now displayed as spider plots (using spider plots, Figs. 4 and 5; pages 6 and 7). Significant differences in ONL thickness and OS lengths were only observed in the 27-week-old Exoc5;Best-1 Cre+ mice cohort (Figures 5C and 5C’; page 7).

Round 2

Reviewer 2 Report

The authors have very satisfactorily addressed my questions and suggestions, including further helpful experimental data. I have no further concerns at this point.